# AdaFPP: Adapt-Focused Bi-Propagating Prototype Learning for Panoramic Activity Recognition

## ABSTRACT

Panoramic Activity Recognition (PAR) aims to identify multi-granularity behaviors performed by multiple persons in panoramic scenes, including individual activities, group activities, and global activities. Previous methods 1) heavily rely on manually annotated detection boxes in training and inference, hindering further practical deployment; or 2) directly employ normal detectors to detect multiple persons with varying size and spatial occlusion in panoramic scenes, blocking the performance gain of PAR. To this end, we consider learning a detector adapting varying-size occluded persons, which is optimized along with the recognition module in the all-in-one framework. Therefore, we propose a novel Adapt-Focused bi-Propagating Prototype learning (AdaFPP) framework to jointly recognize individual, group, and global activities in panoramic activity scenes by learning an adapt-focused detector and multi-granularity prototypes as the pretext tasks in an end-to-end way. Specifically, to accommodate the varying sizes and spatial occlusion of multiple persons in crowed panoramic scenes, we introduce a panoramic adapt-focuser, achieving the size-adapting detection of individuals by comprehensively selecting and performing fine-grained detections on object-dense sub-regions identified through original detections. In addition, to mitigate information loss due to inaccurate individual localizations, we introduce a bi-propagation prototyper that promotes closed-loop interaction and informative consistency across different granularities by facilitating bidirectional information propagation among the individual, group, and global levels. Extensive experiments demonstrate the significant performance of AdaFPP and emphasize its powerful applicability for PAR.

## CCS CONCEPTS

• **Computing methodologies → Activity recognition and understanding**.

## KEYWORDS

Action recognition, Panoramic activity recognition, Prototype learning

ACM MM, 2024, Melbourne, Australia
© 2024 Copyright held by the owner/author(s). Publication rights licensed to ACM.
ACM ISBN 978-x-xxxx-xxxx-x/YY/MM
https://doi.org/10.1145/nnnnnnn.nnnnnnn

**Unpublished working draft. Not for distribution.**

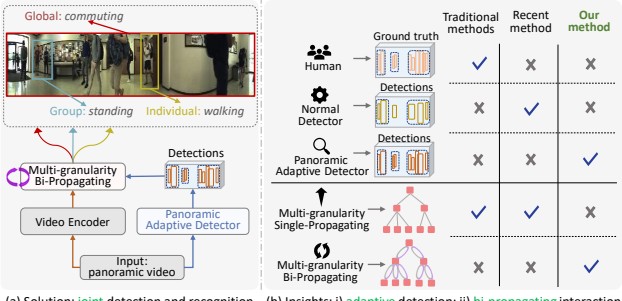

(a) Solution: joint detection and recognition    (b) Insights: i) adaptive detection; ii) bi-propagating interaction

**Figure 1: Our solution and insights. Solution: all-in-one detection and recognition customized for panoramic activities performed by size-varying persons. Insights: i) Adaptive detection instead of ground truth (expensive) and normal detector (for size-similar persons) for size-varying occluded persons; and ii) Bi-propagating interaction instead of single-propagating interaction as the closed-loop interaction for mitigating information loss due to inaccurate localizations.**

## 1 INTRODUCTION

Human activity recognition has garnered significant interest and found extensive applications in diverse fields, such as video surveillance [25, 38] and sports analysis [36, 38]. Over the past decade, researchers have mainly focused on recognizing behaviors at one-single granularity levels, such as individual activities [9, 50], human-human interactions [21, 24], and group activities [23, 47]. The former two commonly pay attention to videos containing only one or a few people, while the latter one focuses on recognizing the overall activity performed by multiple persons. However, some practical scenarios often involve not only unpredictable numbers of individuals but also groups of persons connected with each other through some forms of interaction, e.g., engaging in common activities, which form the additional concepts of group-level activities. For example, in some panoramic scenes within crowded individuals, it is essential to jointly understand individual-level activities and group-level activities.

This work focuses on Panoramic Activity Recognition (PAR) in panoramic scenes, which aims to jointly identify multi-granularity behaviors in crowded panoramic scenes, including individual activities, group activities, and global activities. Unlike normal video scenes in the human activity recognition task, panoramic scenes are characterized by the size-varying occluded persons, as well as multi-granularity activities interacting with each other. Therefore, the key challenges of the PAR task lie in two main aspects: 1) how to accurately detect size-varying persons in crowded scenes; and 2) how to capture the interaction among multi-granularity activities for better recognizing them.

Traditional methods [1, 15] exclusively focus on recognizing multi-granularity activities with the prior of the individual position information. Generally, the solution is to extract individual features based on bounding boxes and then learn the multi-granularity features by modeling the interaction among multiple granularities. For example, Cao et al. [1] proposed to mine intra- and inter-interaction synchronously from individual features for the unified perception across three granularities. However, these above methods relying on manually annotated bounding boxes are not only labor-intensive but also inefficient for real-world deployment. Therefore, recent method [17] attempts to perform individual detection using a normal detector before conducting multi-granularity activity recognition during inference. Nonetheless, normal detectors designed for normal scenes struggle to adapt to panoramic scenes involving multiple persons with varying sizes and spatial occlusion. Moreover, multiple-granularity activities in panoramic scenes mutually interact with each other, thus the information loss caused by inaccurate individual detections may interfere with the performance of multi-granularity activity recognition.

In light of the challenges mentioned above, we attempt to utilize an adaptive detector tailored for panoramic scenes as opposed to the normal detectors. To mitigate information loss due to inaccurate localizations, we further explore enhancing bidirectional multi-granularity information propagation that realizes the closed-loop interaction among multi-granularity activities. To this end, we present a new solution to learn a detector adapting varying-size occluded persons, which is optimized along with the recognition module within the multi-granularity bi-propagating module in an end-to-end way. As shown in Figure 1, such a solution integrating detection and recognition tasks into the all-in-one framework demonstrates two main insights: i) Adaptive detection instead of GT (expensive) and normal detector (for size-similar persons) for size-varying occluded persons; and ii) Bi-propagating interaction instead of single-propagating interaction as the closed-loop interaction for mitigating information loss due to inaccurate localizations.

Formally, we propose a novel Adapt-Focused bi-Propagating Prototype learning (AdaFPP) framework to jointly recognize individual activities, group activities, and global activities in panoramic activity scenes by learning an adapt-focused detector and multi-granularity prototypes as the pretext tasks in an end-to-end way, as shown in Figure 2. Specifically, we design a new Panoramic Adapt-Focuser (PAF) that effectively detects individuals in a coarse-to-fine manner to address the challenges of varying-size and occluded individuals in crowded panoramic scenes. First, we employ a detection network to obtain original detections of individuals. Second, we apply a dense region merging strategy to greedily merge the original detections into dense sub-regions of small-size individuals, which are further cropped and input into the detection network for obtaining the fine-grained detections. Finally, the original and fine-grained detections are fused into the size-adapting detections of individuals. To mitigate the information loss caused by inaccurate localizations in PAF for crowded panoramic activities, we further design a new Bi-Propagation Prototyper (BPP) that models activities at all granularities in a bi-propagative way. First, we encode the panoramic frames to obtain individual features with size-adapting detections. Second, we learn the multi-granularity prototypes from patch embeddings of individual features based on

the hierarchical unified bidirectional encoding blocks, firstly starting with "Individual to Group to Global" propagative interaction, and then starting with "Global to Group to Individual" propagative interaction. Extensive experiments on the dataset are conducted to evaluate the performance of the proposed method.

Overall, the main contributions of this work are summarized as follows,

- **New all-in-one framework**: we propose an end-to-end Adapt-focused bi-Propagating Prototype learning (AdaFPP) framework to jointly recognize individual activities, group activities, and global activities in panoramic activity scenes by learning an adapt-focused detector and multi-granularity prototypes.
- **New varying-size object detector**: To detect the varying sizes and spatial occlusion of multiple persons in panoramic videos, we introduce an effective Panoramic Adapt-Focuser (PAF), achieving size-adapting detection even for small-size individuals by performing fine-grained detections from original detections.
- **New feature learning module**: To mitigate the information loss caused by inaccurate localizations, we introduce a flexible Bi-Propagating Prototyper (BPP) that promotes the closed-loop interaction and informative consistency across multiple granularities by facilitating bidirectional information propagation among individual, group, and global levels.

## 2 RELATED WORK

### 2.1 Multi-person Activity Recognition

Multi-person Activity Recognition focuses on recognizing activities involving multiple people. Most research is currently focused on human-human interaction recognition [21, 24] and group activity recognition (GAR) [27, 33, 52]. The former focuses on recognizing the interactive activity between humans while the latter focuses on recognizing the overall activity of a group of people. Unlike recognizing the action of a single person, graph-based [32] and transformer-based approaches [3] have been widely used to model the spatiotemporal relationship between multiple persons. However, recent works have begun considering a more comprehensive understanding of multi-person activities in crowded scenes [1, 15]. For example, Han et al. [15] proposed to extract individual features based on bounding boxes and then learn the multi-granularity features by modeling the interaction among multiple granularities, including individual, group, and global. Similarly, Cao et al. [1] proposed to mine intra and inter-relevant semantics synchronously from individual features for the unified perception across three granularities. In contrast to these methods based on manually annotated bounding boxes, our work delves into the concurrent tasks of individual detection and multi-person activity recognition.

### 2.2 Spatio-TemporalAction Detection

Spatial-Temporal Action Detection (STAD) aims to localize actions in long untrimmed videos in both spatial and temporal spaces, as well as classify these actions. This is an essential and challenging task in video understanding. Recent research on the STAD task can be mainly categorized into two classes, including two-stage STAD [11, 12, 35, 39, 41] and single-stage STAD [29, 37, 43, 53].

Two-stage STAD relied on off-the-shelf bounding box detections pre-computed at high-resolution videos and proposed transformer models that focus on the recognition task alone. Take Faster RCNN-R101-FPN [34] detector as one example, it is originally trained for human detection on the COCO [16] dataset and subsequently fine-tuned on the AVA [28]. Single-stage STAD achieved both action localization and recognition by sacrificing efficient performance, namely utilizing part of the network to share the majority of the workload. More recent works [4, 53] leveraged recent advancements of DETR [2] in person detection. Unlike the above single-stage STAD works proposed for single-granularity action recognition, we focus on jointly detecting and recognizing multi-granularity actions in crowded panoramic scenes to achieve a more comprehensive understanding of panoramic activities.

## 2.3 Small Object Detection

Compared with normal object detection, small Object detection is more challenging due to the small-size and low-resolution objects in scenes. In addition to the common issues in normal object detection, e.g., object occlusion and inaccurate localizations [5], etc., some remaining issues exist when it comes to small object detection tasks, primarily including object information loss and bounding box perturbation. Previous works [17, 44] mainly adopted uniform segmentation detection, leading to inefficiencies during inference. Recently, some works [6, 20, 31, 46, 49] initially extracted sub-regions containing small objects by utilizing a coarse detector and then employed a fine detector in these regions to detect small objects. Following this paradigm, both Duan et al. [8] and Li et al. [22] exploited pixel-wise supervision for density estimation, achieving more accurate density maps that characterize object distribution well. In this work, we extend this paradigm to the localization of varying-size humans in crowded panoramic scenes. However, due to the knock-on effect of inevitable detection errors on multi-granularity activity recognition, we also compensate for the information loss by cross-granularity bi-propagation to strengthen informative consistency.

## 3 METHODOLOGY

### 3.1 Overview

**Problem Definition.** We assume that one input panoramic video is denoted as $v_p \in \mathbb{R}^{C \times T \times H \times W}$, where $C$ is the number of channels, $T$ is the total number of frames, $H$ and $W$ are the resolution of the frame. The proposed AdaFPP aims to jointly predict its individual activity label, group activity label, and global activity label by learning an adapt-focused detector and multi-granularity prototypes.

**Overall Framwork.** The framework of AdaFPP is shown in Figure 2. On the one hand, the video $v_p$ is fed into the detection branch, where the detection network outputs a set of original detection boxes $B_{ori}$. Subsequently, $B_{ori}$ is input into the Panoramic Adapt-Focuser (PAF) to obtain the ultimate size-adapting detections $B_{ada}$. On the other hand, the video $v_p$ is also fed into the recognition branch, where a pre-trained encoder is used to encode it and obtain the feature map. Following this, we use the size-adapting detections $B_{ada}$ to obtain the feature of each individual. Following ViT [7], we flatten the individual features into $f_p \in \mathbb{R}^{N \times (P^2 \cdot \overline{C})}$, where $\overline{C}$

is the channel, $(P, P)$ is the size of the patch of the individual feature, and $N = \overline{HW}/P^2$ is the number of patches. We project all patches into $D$ dimensions via a linear projection to obtain the patch embeddings $f_{patch} \in \mathbb{R}^{N \times d}$. Subsequently, $f_{patch}$ is input into the Bi-Propagating Prototyper (BPP) to obtain the final feature representations $f_{ind} \in \mathbb{R}^{Q \times d}$, $f_{gro} \in \mathbb{R}^{L \times d}$, and $f_{glo} \in \mathbb{R}^{1 \times d}$ at hierarchical levels for recognition, where $Q$ and $L$ denote the number of individuals, and the number of groups, respectively. Finally, the detection loss $\mathcal{L}_{det}$ and multi-granularity recognition loss $\mathcal{L}_{rec}$ jointly train the whole model.

### 3.2 Panoramic Adapt-Focuser (PAF)

Panoramic Adapt-Focuser (PAF) is designed to address size variation and spatial occlusion issues of individuals in panoramic scenes. Its implementation has four stages: 1) adaptively resizing the bounding boxes of original detections via the Adaptive Object Resizing (AOR) strategy; 2) selecting the dense sub-regions via the Dense Region Merging (DRM) strategy; 3) cropping and inputting the selected dense sub-regions into the Detection Network for obtaining the fine-grained detections; and 4) integrating the original detections and the fine-grained detections to obtain the final size-adapting detections via Detections Fusion strategies. Following are the details in terms of the Adaptive Object Resizing (AOR), Dense Region Merging (DRM), and Detections Fusion strategies.

**Adaptive Object Resizing (AOR).** To mitigate severe biases and heavy overlaps in original detection, we adopt an adaptive object resizing strategy. Specifically, we expand the width and height of each bounding box in the original detection box set $B_{ori}$ from the center with an expansion ratio $\beta$ to enclose its ground truth roughly. Following the small object detection in [18], we control the expansion ratio for different-sized individuals with a threshold $\theta$ in panoramic scenes. This can be expressed as follows:

$$\beta = \begin{cases} \beta_1, & w_h \geq \theta; \\ \beta_2, & \text{otherwise}, \end{cases} \tag{1}$$

where $w_h$ indicates the width of the individual detection box. From this, we obtain the extended detection boxes set, denoted by $B_{ext}$.

**Dense Region Merging (DRM).** We first select the box $a$ with the minimal size from the extended detection box set $B_{ext}$ as the generation starting point. Let $b$ denote one box that belongs in the box set $B_{ext}$ excluding the box $a$, we can obtain the smallest merged box $c$ that encloses the union set $a \cup b$. If the box area of $a \cap b$ is non-zero, namely $a \cap b$ is not $\Phi$, we update $a$ with $c$ and remove $b$ from $B_{ext}$. This process is repeated until $a \cap b$ is $\Phi$. In this case, $a$ as one sub-region is collected into the sub-region set $B_{sub}$. We repeat the above procedure until $B_{ext}$ turns to an empty set, and then obtain one collected dense sub-region set $B_{sub}$.

**Detections Fusion.** Based on sub-regions $B_{sub}$, we crop the corresponding sub-regions from the original frames and input them into the detection network for obtaining fine-grained detections $B_{fin}$. Subsequently, we calculate the final size-adapting detections $B_{ada}$ via Non-Maximum Suppression (NMS) [30], as follows:

$$B_{ada} = \text{NMS}(B_{ori} + B_{fin}), \tag{2}$$

where $\text{NMS}(\cdot)$ indicates the operation of NMS.

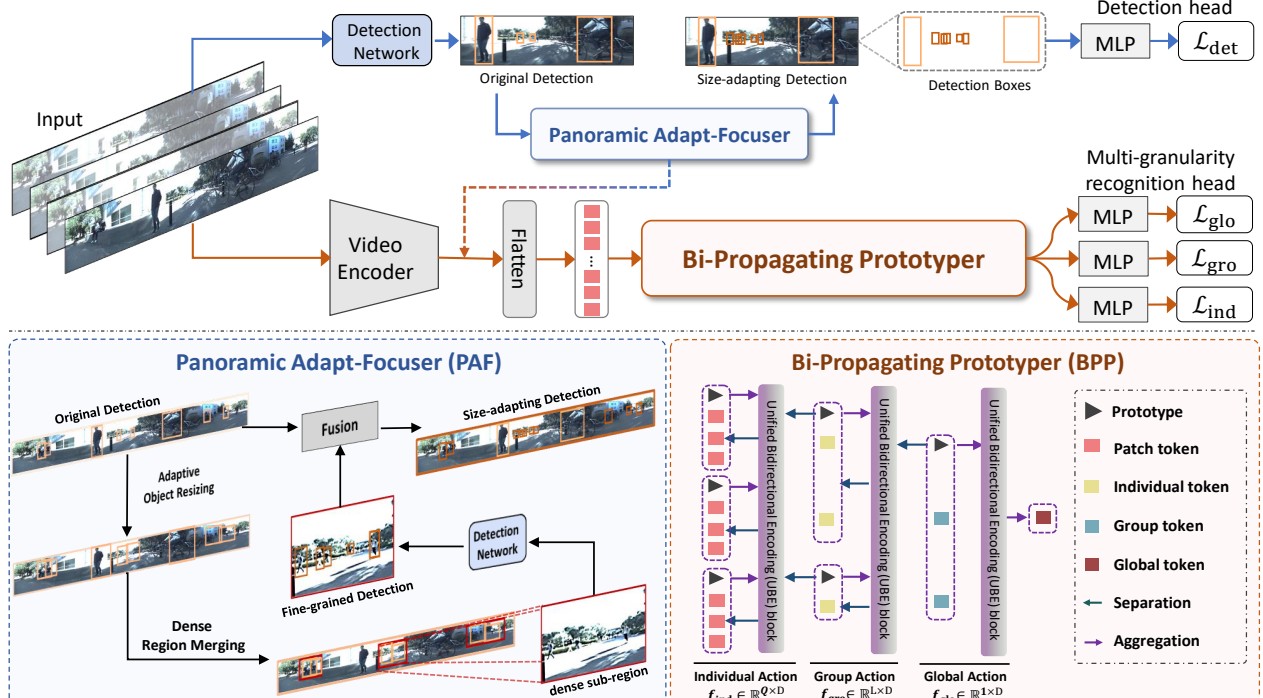

**Figure 2: Framework of the proposed AdaFPP. It consists of two crucial components, i.e., Panoramic Adapt-Focuser (PAF) and Bi-Propagating Prototyper (BPP). PAF comprehensively localizes individuals in crowded panoramic scenes by adaptively selecting and performing fine-grained detections from original detections. BPP learns the multiple-granularity prototypes by prompting the close-loop interaction in a bi-propagatively way. Finally, the detection and recognition heads are jointly used for optimizing the whole model in an end-to-end way.**

### 3.3 Bi-Propagating Prototyper (BPP)

To mitigate the information loss arising from inaccurate localizations in PAF, we introduce a Bi-Propagating Prototyper (BPP) that promotes the closed-loop interaction and informative consistency across multiple granularities by facilitating bidirectional information propagation among the individual, group and global levels. Initially, the forward information propagation is implemented, spanning from individual to group to global levels. Subsequently, the backward information propagation ensues, traversing from global to group to individual levels. Global information is harnessed to guide learning at lower levels in this process, facilitating the informative interaction across multiple granularities. Specifically, BPP is equipped with three Unified Bidirectional Encoding (UBE) blocks, which are detailed in the following.

*3.3.1* **UBE block.** Given the feature sequence $x^l = \{x_i^l \in \mathbb{R}^{1 \times d}\}_{i=1}^M$ as the input of one UBE block, where index $l \in \{\text{patch}, \text{ind}, \text{group}\}$ denotes different granularities and $M$ is the number of tokens, all UBE blocks aim to encode multi-granularity features by learning multi-granularity prototypes $p^l = \{p_j^l \in \mathbb{R}^{1 \times d}\}_{j=1}^J$, where $J$ is the number of tokens. The UBE block is comprised of two parts: UME (Bottom-up Encoding) and CME (Top-down Encoding), as shown in Figure 3.

**UME: Bottom-up Encoding.** To model the interactions within each granularity, we use the Unified Motion Embedding (UME)

module proposed in [15]. As shown on the left side of Figure 3, the learned embedding $x_{\text{cls}}^l \in \mathbb{R}^{1 \times d}$ is prepended to the sequence of $x^l = \{x_i^l\}_{i=1}^M$. We also add the learnable positional embedding $P \in \mathbb{R}^{(M+1) \times d}$ to obtain the input tokens $z^l$. Based on $z^l$, we acquire the output $\hat{z}^l$ by employing the Multi-head Self-Attention (MSA) [40]:

$$z^l = [x_{\text{cls}}^l, x_1^l, x_2^l, ..., x_M^l] + P; \qquad (3)$$

$$\bar{z}^l = \text{MSA}(z^l) + z^l; \qquad (4)$$

$$\hat{z}^l = \text{MLP}(\bar{z}^l) + \bar{z}^l, \qquad (5)$$

where $\text{MLP}(\cdot)$ denotes a multilayer perception consisting of two linear projections.

Moreover, we input feature sequence $\{x_i^l\}_{i=1}^M$ together with learnable prototypes $\{p_j^l\}_{j=1}^J$ to group visual semantics across different granularities. The similarity matrix $A$ between the learnable prototype $p_j^l$ and the token $x_i^l$ can be defined as:

$$A_{i,j} = \frac{\exp(W_q^p p_j^l \cdot W_k x_i^l + \gamma_j)}{\sum_{j'=1}^J \exp(W_q^p p_{j'}^l \cdot W_k x_i^l + \gamma_{j'})}, \qquad (6)$$

where $\gamma_j$ is an independent identically distributed random sample drawn from a $\text{Gumbel}(0, 1)$ distribution [45], $W_q^p$ and $W_k$ are the linear projection weights of the prototype and visual tokens, respectively. After that, we update each prototype $p_j^l$ via aggregating

the feature sequence $x^l$ with different weights into $\hat{p}_j^l \in \mathbb{R}^{1 \times d}$, and then average all prototypes into $o_r^l$, as follows:

$$\hat{p}_j^l = p_j^l + W_o \frac{\sum_{i=1}^{M} A_{i,j} W_v x_i^l}{\sum_{i=1}^{M} A_{i,j}}, \tag{7}$$

$$o_r^l = \text{AvgPool}([\hat{p}_1^l, \cdots, \hat{p}_j^l]); \tag{8}$$

where $W_o$ and $W_v$ are weights used to project and merge features. A regular grid structure does not constrain the block and can reorganize information into arbitrary image fragments. For the bottom-up encoding, we obtain the feature for each level as:

$$\overline{o}^l = o_u^l + o_r^l, \tag{9}$$

where $o_r^l \in \mathbb{R}^{1 \times d}$, and $o_u^l \in \mathbb{R}^{1 \times d}$ is the CLS token from $\hat{z}^l$.

**UBE: Top-down Encoding**. In addition, as shown on the right side of Figure 3, we introduce the reverse Cross-granularity Motion Embedding (CME), aimed at leveraging higher-level information for guiding the learning of low-level features. Specifically, it takes the higher-level output tokens $\overline{o}^{l+1}$ and lower-level visual tokens $x^l$ as inputs to achieve complementary information interaction across different granularities. We obtain the final feature $o^l$ via Multi-Headed Cross-Attention (MCA) [40], as follows:

$$\overline{x}^l = \text{MCA}(x^l, \overline{o}^{l+1}) + x^l; \tag{10}$$

$$o^l = \text{AvgPool}(\text{MLP}(\overline{x}^l) + \overline{x}^l). \tag{11}$$

*3.3.2* **Bi-Propagating with UBE.** Given the patch embeddings $f_{\text{patch}} \in \mathbb{R}^{N \times d}$ from the Video Encoder, we build a bidirectional hierarchical network with the UBE blocks to model the different-granularity activities, which consists of two procedure: forward and backward.

**Forward: Individual→Group→Global**. In UBE, we input the patch embeddings $f_{\text{patch}}$ into its UME module to obtain the individual feature $f_{\text{ind}}' \in \mathbb{R}^{Q \times d}$, group feature $f_{\text{gro}}' \in \mathbb{R}^{L \times d}$ and global feature $f_{\text{glo}} \in \mathbb{R}^{1 \times d}$ across granularity aggregation successively. It can be formulated as:

$$f_{\text{ind}}' = \text{UME}_{\text{p2i}}(f_{\text{patch}}); \tag{12}$$

$$f_{\text{gro}}' = \text{UME}_{\text{i2g}}(f_{\text{ind}}'); \tag{13}$$

$$f_{\text{glo}} = \text{UME}_{\text{g2g}}(f_{\text{gro}}'), \tag{14}$$

where the subscript of UME indicates the input and output of different granularities.

**Backward: Group←Global**. As shown in Figure 3, we input the global feature $f_{\text{glo}}$ obtained at the global level together with the visual tokens at the group level into the CME to acquire the final group representation $f_{\text{gro}} \in \mathbb{R}^{L \times d}$. It can be formulated as:

$$f_{\text{gro}} = \text{CME}_{\text{g2g}}(f_{\text{glo}}), \tag{15}$$

where $\text{CME}_{\text{g2g}}$ aims to utilize global information to guide group-level representation learning.

**Backward: Individual←Group**. Similarly, we input the feature representation $f_{\text{gro}}$ obtained at the group level together with the visual tokens at the individual level into the CME to acquire the ultimate individual representation $f_{\text{ind}} \in \mathbb{R}^{Q \times d}$, as follows:

$$\begin{aligned} f_{\text{ind}} &= \text{CME}_{\text{g2i}}(\text{CME}_{\text{g2g}}(f_{\text{glo}})) \\ &= \text{CME}_{\text{g2i}}(f_{\text{gro}}), \end{aligned} \tag{16}$$

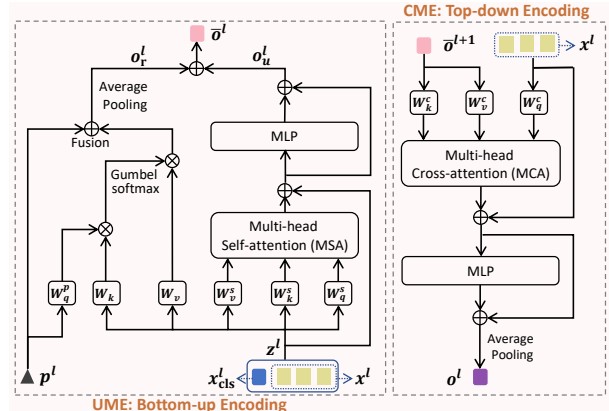

**Figure 3: Detailed architecture of one Unified Bidirectional Encoding (UBE) block. It includes the UME (bottom-up encoding) and CME (top-down encoding) modules. $l$ is defined as the $l \in \{patch, ind, group\}$, and $l+1$ is the higher-level granularity of $l$.**

where $\text{CME}_{\text{g2i}}$ aims to utilize group information to guide individual-level representation learning.

## 3.4 Overall Optimization

The overall training loss of the AdaFPP combines the conventional detection loss $\mathcal{L}_{\text{det}}$ in Eq. 17 and the multi-granularity recognition loss $\mathcal{L}_{\text{rec}}$ in Eq. 18. Following [14], the detection loss is defined as:

$$\mathcal{L}_{\text{det}} = \lambda_{\text{reg}} \mathcal{L}_{\text{reg}} + \mathcal{L}_{\text{obj}} + \mathcal{L}_{\text{cls}}, \tag{17}$$

where $\mathcal{L}_{\text{reg}}$ denotes the IoU loss for the regression loss of the bounding boxes, $\mathcal{L}_{\text{obj}}$ denotes the cross-entropy loss over two classes, and $\mathcal{L}_{\text{cls}}$ denotes the cross-entropy loss used for classification. $\lambda_{\text{reg}}$ is the constant scalar balancing the contributions of the loss term, whose default value is 5.

For the recognition loss, we use the same multi-granularity loss as JRDB-PAR [15] for panoramic activity recognition. The recognition loss is defined as:

$$\mathcal{L}_{\text{rec}} = \mathcal{L}_{\text{i}} + \mathcal{L}_{\text{s}} + \mathcal{L}_{\text{g}} + \mathcal{L}_{\text{d}}, \tag{18}$$

where $\mathcal{L}_{\text{i}}$, $\mathcal{L}_{\text{s}}$, $\mathcal{L}_{\text{g}}$, and $\mathcal{L}_{\text{d}}$ denote the binary cross-entropy loss function for the individual, group, global activity recognitions, as well as the group detection.

Finally, the overall training loss for the proposed AdaFPP is formulated as follows:

$$\mathcal{L} = \mathcal{L}_{\text{rec}} + \lambda \mathcal{L}_{\text{det}}, \tag{19}$$

where $\lambda$ is the weight coefficient.

# 4 EXPERIMENTS

## 4.1 Dataset

We evaluate the proposed method on the challenging Panoramic Activity Recognition benchmark: JRDB-PAR [15]. It is based on JRDB [26] and JRDB-Act [10] datasets, which include 360° RGB videos of crowded multi-person scenes captured by a mobile robot. The dataset provides individual detection boxes with IDs, individual activities, group-level detections, as well as manually annotated

**Table 1: Comparative performance (%) of panoramic activity recognition.** Results (marked gray color) with the help of expensive GT detection information in training and inference are only regarded as the reference. The superscript * denotes that we reproduce results by using ground-truth group detection instead of group-level detections.

| Extra | Method | Individual Activity | | | Group Activity | | | Global Activity | | | Overall |
|---|---|---|---|---|---|---|---|---|---|---|---|
| | | $P_i$ | $R_i$ | $F_i$ | $P_p$ | $R_p$ | $F_p$ | $P_g$ | $R_g$ | $F_g$ | $F_a$ |
| Ground Truth | AT* [13] | 65.6 | 54.5 | 57.0 | 28.3 | 26.2 | 26.8 | 25.3 | 20.3 | 21.9 | 35.2 |
| | HIGCIN* [48] | 16.5 | 13.1 | 14.0 | 16.8 | 15.3 | 15.4 | 71.7 | 47.4 | 55.2 | 28.2 |
| | Dynamic* [51] | 62.2 | 66.9 | 60.3 | 38.6 | 39.2 | 37.9 | 25.3 | 20.6 | 22.2 | 40.1 |
| | ARG [42] | 27.7 | 21.6 | 23.4 | 12.1 | 11.3 | 11.5 | 66.7 | 45.6 | 52.6 | 29.1 |
| | JRDB-PAR [15] | 34.8 | 26.9 | 29.1 | 14.1 | 13.7 | 13.8 | 78.3 | 46.8 | 57.3 | 33.4 |
| | MUP [1] | 71.0 | 58.3 | 61.5 | 28.1 | 30.0 | 28.0 | 52.8 | 44.8 | 47.3 | 45.6 |
| Detector | AT [13] | 39.9 | 33.8 | 34.7 | 10.3 | 10.0 | 10.1 | 36.0 | 24.7 | 28.5 | 24.4 |
| | HIGCIN [48] | 30.8 | 21.2 | 24.0 | 12.3 | 11.8 | 11.9 | 16.4 | 14.5 | 15.1 | 17.0 |
| | Dynamic [51] | 49.0 | 47.2 | 44.9 | 10.7 | 9.3 | 9.8 | 25.2 | 20.1 | 21.7 | 25.4 |
| | ARG [42] | 24.3 | 31.4 | 21.4 | 15.7 | 15.1 | 15.2 | 24.1 | 21.4 | 19.6 | 20.2 |
| | JRDB-PAR [15] | 23.8 | 18.7 | 20.0 | 18.5 | 23.7 | 19.6 | **60.5** | 33.2 | 42.3 | 27.3 |
| | MUP [1] | 48.6 | 36.1 | 39.1 | 20.2 | 25.5 | 21.6 | 56.4 | 39.4 | 45.1 | 35.3 |
| | AdaFPP (**Ours**) | **63.8**(+14.8) | **53.3**(+6.1) | **54.5**(+9.6) | **25.8**(+5.6) | **31.3**(+5.8) | **26.7**(+5.1) | 55.7(-4.8) | **42.8**(+3.4) | **47.1**(+2.0) | **42.8**(+7.5) |

group activities and global activities. Specifically, the dataset contains 27 videos (20/7 for training/testing) with the keyframes (one keyframe in every 15 frames) selected for annotation and evaluation. This dataset consists of 27,920 frames with over 628k human bounding boxes. The categories of individual activity, group activity, and global activity are 27, 11, and 7, respectively.

### 4.2 Protocol

We adopt the same metrics in JRDB-PAR [15], including precision, recall, and $F_1$ score. However, due to the joint tasks of individual detection and activity recognition, slight modifications were made to these metrics, as detailed below:

**Protocol on Individual Activity.** Individual Activity recognition includes individual detection and action recognition. For the individual detection, the IoU > 0.3 between the detected box and ground-truth bounding box is taken as the true detected individual. We further consider the action recognition result. The true detected individuals with the correct action prediction are taken as the true individual's action predictions. After that, we denote the precision, recall, and $F_1$ score as $P_i$, $R_i$, and $F_i$, respectively.

**Protocol on Group Activity.** Similar to the individual, we first adopt the generic protocol from the group detection task [15]. i.e., the predicted group members are treated as the final predicted group only if their IoU > 0.3 concerning the real group members. After getting the predicted group, we perform group activity recognition. We calculate the precision, recall, and $F_1$ score as $P_p$, $R_p$, and $F_p$, respectively.

**Protocol on Global Activity.** We also denote the precision, recall and $F_1$ score as $P_g$, $R_g$ and $F_g$ for the global activity recognition.

**Protocol on Overall Activities.** The overall metric for the panoramic activity detection task is the average value of $F_i$, $F_p$, and $F_g$ covering three granularities, which is defined as $F_a = Average(F_i + F_p + F_g)$.

### 4.3 Setting

In experiments, a total of 1,439 keyframes are used for training, while 411 key frames are reserved for testing, where the size of each frame is 480×3, 760 in default. We use the Adam [19] optimizer with the initial learning rate of $2 \times 10^{-5}$ and the weight decay of $10^{-2}$. Mini-batch size is set to 4, and the training epoch is set to 40 with the learning rate is decayed after 30 epochs. We adopt Yolox [14] as the Detection Network for all methods when requiring the detector. By default, we set the parameter $\theta$ in Eq. (1) to 48, parameters $\beta_1$, and $\beta_2$ in Eq. (1) to 1.5, and 1.8, as well as the parameter $\lambda$ in Eq. (19) to $1\times10^{-3}$. The implementation of the overall framework is carried out on PyTorch in a Linux environment with an NVIDIA GeForce 3090. Additionally, we report FLOPs and Params of the proposed method and some comparative methods in the supplementary material due to space limitation.

### 4.4 Comparison with State-of-the-arts

Panoramic Activity Recognition (PAR) involving three-granularity activity recognitions is a novel and challenging task, as there are few directly comparative methods available. We evaluate the proposed AdaFPP on the JRDB-PAR [15] dataset by comparing it with current representative methods, including the benchmark methods JRDB-PAR [15] and MUP [1]. In addition, we also compare against several state-of-the-art methods on individual activities and group activity recognition, e.g., AT [13], HIGCIN [48], Dynamic [51], and ARG [42]. We make appropriate adjustments to these comparative methods for adapting to the three-granularity activity recognitions existing in PAR. Table 1 shows the comparative performance of PAR obtained by different methods. First, when using the extra ground-truth detections, we report the performance (marked with gray color in Table) obtained by the comparative methods. Since

**Table 2: A set of ablation studies for the proposed method.**

| PAF | BPP | $F_i$ | $F_p$ | $F_g$ | $F_a$ |
|---|---|---|---|---|---|
| ✗ | ✗ | 39.1 | 21.6 | 45.1 | 35.3 |
| ✓ | ✗ | 49.1 | 23.8 | 44.2 | 39.1 |
| ✗ | ✓ | 40.2 | 21.1 | 49.2 | 36.9 |
| ✓ | ✓ | 54.5 | 26.7 | 47.1 | 42.8 |

(a) Ablation studies on each component.

| Input | $F_i$ | $F_p$ | $F_g$ | $F_a$ |
|---|---|---|---|---|
| $160 \times 1,250$ | 38.7 | 24.2 | 51.4 | 38.1 |
| $240 \times 1,880$ | 43.3 | 22.0 | 52.2 | 39.2 |
| $480 \times 3,760$ | 54.5 | 26.7 | 47.1 | 42.8 |

(b) Ablation studies on input resolution.

| Method | detect | $F_i$ | $F_p$ | $F_g$ | $F_a$ |
|---|---|---|---|---|---|
| PAR [15] | w/o Ada | 23.2 | 23.0 | 36.9 | 27.8 |
| | Ada | 28.6 | 16.8 | 50.9 | 32.1 |
| MUP [1] | w/o Ada | 41.8 | 19.8 | 45.7 | 35.4 |
| | Ada | 49.1 | 23.8 | 44.2 | 39.1 |
| Ours | w/o Ada | 38.8 | 21.1 | 52.5 | 37.4 |
| | Ada | 54.5 | 26.7 | 47.1 | 42.8 |

(c) Ablation studies on PAF.

| $\beta_1$ | $\beta_2$ | $F_i$ | $F_p$ | $F_g$ | $F_a$ |
|---|---|---|---|---|---|
| 1.2 | 1.2 | 45.0 | 22.4 | 45.7 | 37.7 |
| 1.2 | 1.5 | 39.8 | 26.7 | 51.2 | 39.2 |
| 1.5 | 1.8 | 54.5 | 26.7 | 47.1 | 42.8 |
| 1.8 | 2.0 | 49.3 | 27.1 | 41.6 | 39.3 |
| 1.8 | 2.3 | 47.9 | 26.9 | 52.9 | 42.6 |
| 2.0 | 2.3 | 43.4 | 19.9 | 51.8 | 38.4 |

(d) Ablation studies on expansion ratio.

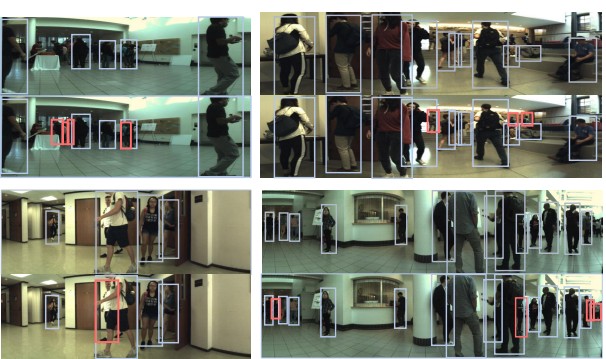

**Figure 4: Visualization of individual detections by PAF. The gray boxes indicate the original detections, and the red boxes indicate the additional fine-grained detections by PAF.**

these methods directly use manually-annotated bounding boxes, these performance results are only retarded as the reference. It is noted that AdaFPP without using ground-truth detections achieves $F_a$ of 42.8%, which is comparable to $F_a$ of 45.6% achieved by the SOTA method using ground-truth detections. This demonstrates the effectiveness of PAF of AdaFPP for detecting panoramic activity. Second, when using an extra detector for individual detection, we employ the clustering algorithm to obtain detections of groups by grouping several individuals into one group. Compared with the other method using extra detector, AdaFPP achieves a state-of-the-art performance (i.e., $F_a$ of 42.8%) on the overall score. In particular, AdaFPP performs best on all protocols of individual, group, and global levels, except for the $P_g$. This demonstrates the effectiveness of the proposed framework for recognizing panoramic activity.

## 4.5 Ablation Studies

**Effectiveness of PAF and BPP.** Table 2a summarizes the effectiveness of each component. Specifically, compared with the baseline, PAF (Panoramic Adapt-Focuser) has shown performance improvements of 10% (on $F_i$) at the individual level, 2.2% (on $F_p$) at the

group level, and an overall improvement of 3.8% (on $F_a$). These improvement gains demonstrate that PAF can more accurately detect individuals in panoramic scenes, thereby facilitating individual and group recognition. Moreover, using BPP (Bi-Propagating Prototyper) gains a little improvement of 1.6% (on $F_a$) in overall performance. However, when combining PAF and BPP, the performance improvements are significant, i.e., 15.4%, 5.1%, and 2.0% at the individual, group, and global levels, respectively, leading to an overall performance increase of 7.5%. This indicates that the proposed BPP can effectively integrate with the PAF, mitigating the impact of inaccurate localizations on detection and further enhancing recognition performance.

**Effect of Different Input Resolutions.** To explore the effect of resolution in PAR, we conduct experiments of AdaFPP with frame inputs of varied resolutions. The original video resolution was $480 \times 3,760$, and we reduced it by half and two-thirds compression to obtain the different resolutions, which are input adaFPP. The results are shown in Table 2b. It indicates that the decrease in frame resolution leads to a decline in overall performance. Obviously, high resolution is a benefit for PAR, where the small-scale individuals can be modeled well.

**Superiority of Panoramic Adapt-Focuser (PAF).** To test the superiority of the designed PAF for PAR, we conduct experiments to compare the performance of the normal detector (designed for normal scenes) and PAF (adaptive for panoramic scenes). Here, the representative methods including JRDB-PAR [15] and MUP [1] are employed as the comparative methods. The comparison results are shown in Table 2c, where "w/o Ada" and "Ada" denote the method using a normal detector (e.g., YOLOX [14]) and PAF, respectively. For each method, its performance is improved significantly when equipped with PAF instead of a normal detector. Specifically, JRDB-PAR, MUP, and AdaFPP achieve performance improvements of 4.3%, 3.7%, and 5.4% on $F_a$, respectively, It is well illustrated the superiority of PAF for detecting individuals in the crowed panoramic scenes. Moreover, when using PAR, the performance gain of AdaFPP is more than that of alternatives. which demonstrates PAF is more beneficial to the AdaFPP framework.

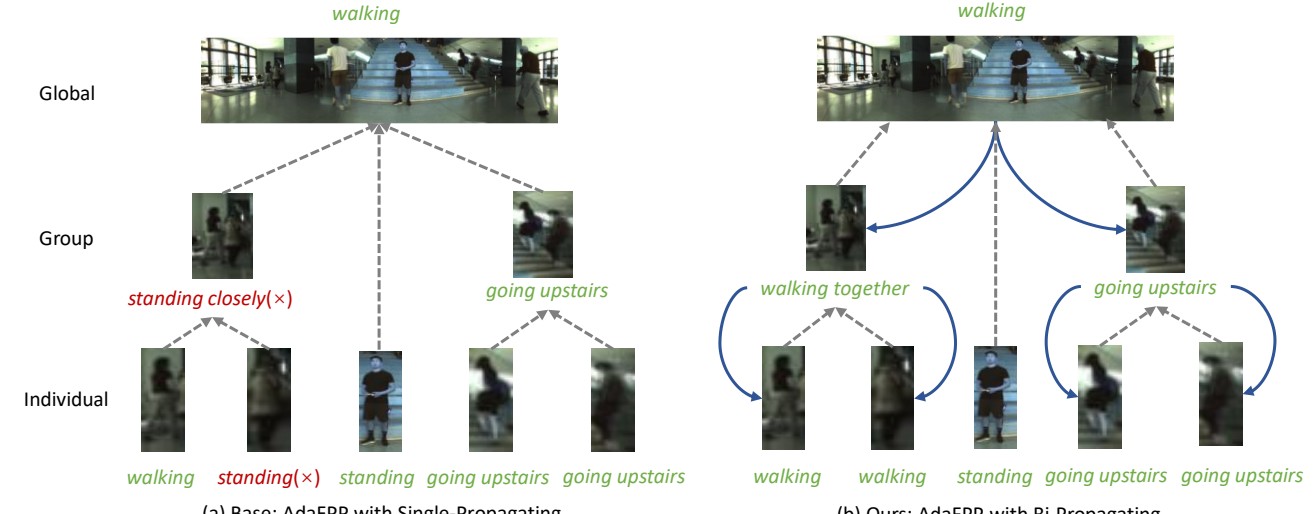

**Figure 5: Comparison visualization at three-granularity activity recognition. Incorrect recognition results are marked in red. (a) has only the bottom-up propagation. (b) has the bottom-up propagation and top-down propagation.**

**Effect of Expansion Ratio in PAF.** The expansion ratio $\beta$ (Eq.1) in PAF is important for detecting individuals. To mitigate severe biases and heavy overlaps of original detection, original detection boxes larger than the threshold $\theta$ are appropriately extended by using the ratio $\beta = \beta_1$, while those smaller than the threshold are appropriately extended by using the ratio $\beta = \beta_2$. The performance obtained by AdaFPP with different values of $\beta_1$ and $\beta_2$ are shown in Table 2d. It can be observed that selecting larger or smaller values of ratios $\beta_1$ and $\beta_2$ may affect the final recognition performance of AdaFPP. We can set the moderate values for $\beta_1$ and $\beta_2$ to obtain the best performance, namely $\beta_1 = 1.5$, and $\beta_2 = 1.8$. Here, the achieved performance of AdaFPP is also better than that of all comparative methods, even if we set any values for $\beta_1 \in [1.2, 2.0]$, and $\beta_2 \in [1.2, 2.3]$.

### 4.6 Qualitative Analysis

**Detection Results of Panoramic Adapt-Focuser.** We investigate the effectiveness of the designed Panoramic Adapt-Focuser (PAF) by visualizing the original detections (using YOLOX [14]) and size-adapting detections (using PAF) within the identical frame from the JRDB-PAR dataset [15]. Here, we also adopt YOLOX as the detection network in PAF for fair comparison. The detection comparison is shown in Figure 4. The gray boxes denote the original detections of individuals by YOLOX, and the red boxes denote the ultimate size-adapting detections by PAF. It can be seen that PAF enables the successful detection of partially occluded and size-small individuals. This demonstrates better applicability of PAF in terms of detection for panoramic scenes.

**Recognition Results with Bi-Propagating.** To investigate the superiority of the proposed BPP, we compare the visualized results of multi-granularity activity recognition obtained by AdaFPP with Single-Propagating and AdaFPP with Bi-Propagating, as shown in Fig. 5. Here, AdaFPP with Single-Propagating means that it uses

Single-Propagating Prototyper instead of Bi-Propagating Prototyper in AdaFPP. It is noted that AdaFPP with Single-Propagating in Fig. 5 (a) inaccurately recognizes one individual activity of "walking" as "standing", and then mistakenly associates the group activity with "standing closely," owing to its single information propagation from individuals to groups. In contrast, AdaFPP with Bi-Propagating in Fig. 5 (b) demonstrates that our introduced bidirectional propagation leverages global "walking" features to steer the reverse process from each group to each individual, in conjunction with forward information propagation, thereby achieving accurate recognition of both individual activities and group activities.

## 5 CONCLUSION

In this work, we propose an end-to-end Adapt-Focused Bi-Propagating Prototype Learning (AdaFPP) framework to jointly recognize individual, group, and global activities in crowed panoramic scenes by learning an adapt-focused detector and multi-granularity prototypes. Overall, AdaFPP has two main insightful components, i.e., Panoramic Adapt-Focuser (PAF), and Bi-Propagating Prototyper (BPP). PAF can adaptively detect multiple persons with varying sizes and spatial occlusion in panoramic scenes. BPP can promote closed-loop interaction and informative consistency across different granularities via bidirectional information propagation among individual, group, and global levels. Extensive experiments on the public dataset validate the effectiveness of the proposed method. However, the PAR-related dataset is not extensive, which makes it difficult to conduct a more comprehensive evaluation of AdaFPP. In the future, we will explore additional panoramic datasets, and investigate the lightweight version of AdaFPP for deploying in real-world scenes.

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
