# OpenReview forum: "AdaFPP: Adapt-Focused Bi-Propagating Prototype Learning for Panoramic Activity Recognition"
_acmmm.org/ACMMM/2024/Conference — MM2024 Poster_

### Official Review · Reviewer_HD3i · 2024-05-20

**Rating:** 4
**Confidence:** 3

**Summary:**

This paper presents a novel framework for Panoramic Activity Recognition (PAR). The authors address the challenges of detecting and recognizing activities of varying sizes and occluded persons in panoramic scenes. The proposed AdaFPP framework integrates detection and recognition tasks in an end-to-end manner, focusing on adaptive detection and bi-propagating information propagation across multiple activity granularities. Specifically, the authors propose a Panoramic Adapt-Focuser (PAF) module that significantly enhances detection accuracy through an adaptive approach, overcoming a series of issues associated with directly applying a norm detector to panoramic images. Subsequently, the authors innovatively introduce a Bi-Propagating Prototyper (BPP) module to enhance information interaction and informative consistency across multiple granularities. Extensive experiments have demonstrated the effectiveness of the methods proposed in this paper.

**Strengths:**

1. The introduction of the Panoramic Adapt-Focuser (PAF) allows for size-adaptive detection, addressing the challenge of varying sizes and occlusions of persons in panoramic scenes
2. The bi-propagation mechanism ensures robust information sharing across different granularities (individual, group, and global levels), which enhances the accuracy of activity recognition
3. Extensive experiments demonstrate the significant performance improvements and applicability of the proposed method for PAR
4. Figures and examples are informative and help understand the paper

**Limitations:**

1. The proposed framework is quite complex, involving multiple components and steps which might be challenging to implement and optimize in real-world scenarios
2. In the PAF module, theta is a crucial parameter for the subsequent DRM. Moreover, different values of theta can significantly affect the division of the final sub-region set. This parameter is likely to have a substantial impact on the final detection results; however, the authors lack relevant analysis and discussion on this parameter
3. Despite the PAF module demonstrating good performance, it lacks novelty. Intuitively, PAF directly segments the panoramic image using hyperparameters to adapt to the resolution of the norm detector for better detection results

**Suitability:**

2

---

### Official Review · Reviewer_tqnA · 2024-05-27

**Rating:** 4
**Confidence:** 2

**Summary:**

In this paper, the author(s) propose an end-to-end framework to jointly recognize individual activities, group activities, and global activities in panoramic activity scenes by learning an adapt-focused detector and multi-granularity prototypes. The method mainly consist of (1) an effective Panoramic Adapt-Focuser to detect the varying sizes and spatial occlusion of multiple persons in panoramic videos, and (2) a flexible Bi-Propagating Prototyper to mitigate the information loss caused by inaccurate localizations.

**Strengths:**

1. Well motivation. It is an effective way to improve the performance of the task to fully consider the varying-size objects in the video to obtain fine-grained individual/group/global representations.

2. Good performance. From the section of experiments, we can see that the proposed method obtain an obvious improvement than previous end-to-end panoramic human activity recognition methods.

**Limitations:**

1. The innovation of the method is a little weak.
    - the detection method in proposed in [1] and the embedding block is partial proposed in [2].
    - in the sections of abstract and introduction, author(s) spend a lot of time explaining that it is difficult for current methods to detect multi-scale person, which is the main difficulty of the  panoramic human activity recognition task. However, the solution is to adopt the method proposed in [2].
     Thus, in my opinion, the main innovation of the paper is actually to propose a bidirectional feature fusion technology.
2. Some experimental results should be added to show the effectiveness of the proposed method comprehensively.
    - In Table 2(c), I think there should present the results of using ground truth bounding boxes of persons.
    - The performance of `Single-Propagating' should also be presented.
3. The description of the proposed method is somewhat difficult to understand. It should be helpful to give an algorithmic pipeline for this method.
4. There are some typos that seriously affect the understanding of the method details. For example, I think Line496 should be CME: Top-down Encoding.

5. The author(s) claim that the varying-size persons in the video cause the low performance of previous methods: "Unlike normal video
scenes in the human activity recognition task, panoramic scenes are characterized by the size-varying occluded persons". But, is there any literature or analysis that supports this opinion? Can we provide an analysis that shows how many small objects in the videos and what would happen to the performance of the method if these small targets are not taken into account? Of course, I think this analysis is a useful support for the methods presented in this paper, but it is only a suggestion and is not required..



[1] Panoramic human activity recognition.

[2] UFPMP-Det: Toward accurate and efficient object detection on drone imagery

**Suitability:**

2

---

### Official Review · Reviewer_ARdi · 2024-06-03

**Rating:** 4
**Confidence:** 2

**Summary:**

This paper presents an end-to-end panoramic activity recognition framework, which introduces an adapt-focused and bi-propagating strategies for model optimization and performs recognition of individual activities, group
activities, and global activities at the same time. This work conducts extensive experiments to verify the effectiveness.

**Strengths:**

1. This paper is well written and easy to follow.
2. This paper presents an end-to-end panoramic activity recognition framework, which combines the recognition tasks of individual activities, group activities, and global activities .
3. The paper introduces an effective Panoramic Adapt-Focuser module for small object detection, and present a Bi-Propagating Prototyper (BPP) for multi-task learning.
4. This work conducts extensive experiments and obtain SOTA performance, which verify the superiority of the proposed method.

**Limitations:**

1. The compared methods are not latest, this paper should compare with a. REACT: Recognize Every Action Everywhere All At Once(2023), b.Spatio-Temporal Proximity-Aware Dual-Path Model for Panoramic Activity Recognition(2024).
2. For Ada module, this paper should compare with more  detection methods for small objects.
3. For BPP ablation study, this paper should provide the detail of baseline without BPP.

**Suitability:**

2

---

### Meta-Review · Area_Chair_48ja · 2024-07-02

**Recommendation:** Accept (Poster)
**Confidence:** 4

**Metareview:**

1. This paper presents an end-to-end panoramic activity recognition framework that jointly recognizes individual activities, group activities, and global activities in panoramic scenes by learning an adapt-focused detector and multi-granularity prototypes.
Specifically, the paper proposes a Panoramic Adapt-Focuser (PAF) module, which significantly enhances detection accuracy through an adaptive approach, overcoming issues associated with directly applying a standard detector to panoramic images.
Additionally, this work introduces a Bi-Propagating Prototyper (BPP) module to enhance information interaction and consistency across multiple granularities.
From the experiments section, it is evident that the proposed method achieves a notable improvement over previous end-to-end panoramic human activity recognition methods.
The paper is well-written, and the figures and examples are informative, aiding in the understanding of the method.

2. However, the innovation of the method is somewhat limited. The Panoramic Adapt-Focuser (PAF) module is designed to detect varying-sized and occluded individuals,
segmenting the panoramic image using hyperparameters to adapt to the resolution of the standard detector for better results, but without much novelty.
Additionally, there are not enough comparisons between the Ada module and other optimized methods for small objects.

Based on these reasons and all reviews rate borderline accept, I tend to accept this paper.